# A Systematic Method for the Identification of Aporphine Alkaloid Constituents in *Sabia schumanniana* Diels Using UHPLC-Q-Exactive Orbitrap/Mass Spectrometry

**DOI:** 10.3390/molecules27217643

**Published:** 2022-11-07

**Authors:** Shuai E, Zi-Chao Shang, Shi-han Qin, Kai-lin Li, Yan-nan Liu, Ji-Li Wu, Fang Yan, Wei Cai

**Affiliations:** 1School of Pharmacy, Weifang Medical University, Weifang 261000, China; 2Hunan Province Key Laboratory for Antibody-Based Drug and Intelligent Delivery System, School of Pharmaceutical Sciences, Hunan University of Medicine, Huaihua 418000, China; 3Nursing School, Hunan University of Medicine, Huaihua 418000, China

**Keywords:** *Sabia schumanniana* Diels, aporphine alkaloids, UHPLC-Q-Exactive Orbitrap MS, neutral loss, diagnostic fragmentation ions, magnoflorine, lirinidine

## Abstract

*Sabia schumanniana* Diels (SSD) is a plant whose stems are used in traditional folk medicine for the treatment of lumbago and arthralgia. Previous studies have revealed chemical constituents of SSD, including triterpenoids and aporphine alkaloids. Aporphine alkaloids contain a variety of active components, which might facilitate the effective treatment of lumbago and arthralgia. However, only 5-oxoaporphine (fuseine) has been discovered in SSD to date. In this study, we sought to systematically identify the aporphine alkaloids in SSD. We established a fast and reliable method for the detection and identification of these aporphine alkaloids based on ultra-high-performance liquid chromatography (UHPLC)-Q-Exactive-Orbitrap/mass spectrometry combined with parallel reaction monitoring (PRM). We separated all of the analyzed samples using a Thermo Scientific Hypersil GOLD™ aQ C18 column (100 mm × 2.1 mm, 1.9 μm). Finally, we identified a total of 70 compounds by using data such as retention times and diagnostic ions. No fewer than 69 of these SSD aporphine alkaloids have been reported here for the first time. These findings may assist in future studies concerning this plant and will ultimately contribute to the research and development of new drugs.

## 1. Introduction

*Sabia schumanniana* Diels (SSD) is a deciduous climbing woody vine of the genus *Sabia* in the family Sabiaceae and is widely distributed in the Sichuan and Guizhou provinces of China. The stems of SSD are used in traditional folk medicine for the treatment of lumbago and arthralgia [1,2]. The main active constituents in the genus *Sabia* are alkaloids [3]; however, only triterpenoids and 5- oxoaporphine (fuseine) have been identified in *Sabia schumanniana* Diels before now [4]. Aporphine alkaloids are natural chemical compounds that are highly biologically active and play an important role in plants. In recent studies, aporphine alkaloids have been shown to exhibit potent anti-diabetic, anti-cancer [5], anti-inflammatory [6], and antivirus properties [7]. Further studies on the aporphine alkaloid components of SSD are therefore warranted.

Ultra-high-performance liquid chromatography-Q-Exactive Orbitrap/mass spectrometry is a process which can be used for chemical constitution identification and offers high selectivity, high sensitivity, and high efficiency [8,9,10]. The fragment information obtained through MS combined with advanced post-processing technology data enables the determination of the diagnostic fragment ions and neutral losses. Typically, sample data acquisition involves a full scan with data-dependent MS^2^ (full MS/dd-MS^2^). However, MS^2^ data cannot be detected in this mode if the relative abundance of MS^1^ ions does not reach a required level. As a result, any desired compounds that are present only in trace amounts are disregarded because of the limitations of the analytical method. Recently, this tool has been used to conveniently acquire MS^2^ data using the parallel reaction monitoring (PRM) detection mode, which allows for the isolation of the targeted precursor ions and product fragment ions from the precursor and enables the detection of the resulting product ions based on the preset isolation window width and collision energy, eliminating most interference. By such means, researchers have achieved the accurate detection and quantification of confirmed and targeted fragments [11,12].

In this study, we systematically characterized SSD constituents using UHPLC-Q-Exactive Orbitrap MS combined with PRM. We putatively identified 70 aporphine alkaloids based on their precise mass measurement, chromatographic retention, MS^n^ spectra analysis, and bibliographical data. No fewer than 69 of these aporphine alkaloids were identified in SSD for the first time in this study. These results may contribute to a better understanding of the medicinal effects of SSD and help to lay the groundwork for the future quality control of SSD-derived medicines in a clinical setting.

## 2. Results and Discussion

From the experimental data for our SSD sample and a summarized fragmentation pattern, we identified a total of 70 aporphine alkaloids. Table 1 and Appendix A give the chromatographic and mass data for these detected constituents and includes retention times (tR), experimental masses, and the discrepancies between the theoretical and experimental masses (in ppm), in addition to molecular formulas for all the aporphine alkaloids as well as MS/MS fragment ions. Figure 1 illustrates the high-resolution extracted ion chromatogram from the SSD extract in the positive ion mode. All compounds are numbered according to their order of elution.

### 2.1. Establishment of the Analytical Method

For this study, we established an analytical strategy based on utilizing UHPLC-Q-Exactive Orbitrap MS combined with parallel reaction monitoring (PRM) to identify diagnostic fragment ions (DFIs) and neutral losses (NLs) in order to comprehensively screen for and detect the aporphine alkaloids present in SSD. First, we injected SSD samples into a UHPLC-Q-Exactive Orbitrap MS to obtain full mass raw data via use of the full-mass scanning mode. Second, we predicted the potential chemical compounds using Compound Discoverer 3.0 and Metabolite Workflow. We determined parameters in line with [13]. The drug was set to magnoflorine, while roemerine and the added group were assigned to a list of substituents including -CH_3_, -OH, -OCH_3_, C=O, and -OCH_2_O-. Third, we collected fragmentation ions using UHPLC-Q-Exactive Orbitrap MS based on the parallel reaction monitoring mode activated by inclusion ions from the list described above. Finally, we performed an accurate full-scan mass spectrometry and MS^2^. We also extracted the retention time information and incorporated relevant database and literature data. By such means, we obtained our SSD identification results.

### 2.2. Identifification and Analysis of Aporphine Alkaloids in SSD

We used UHPLC-Q-Exactive Orbitrap MS to examine the fragmentation patterns of four reference standards in positive mode in order to establish the neutral loss and the diagnoses for fragmentation ions.

Figure 2A shows the proposed fragmentation pathway for magnoflorine. This generated a fragment ion at *m*/*z* 297.1123 (C_18_H_17_O_4_^+^) via the neutral loss at *m*/*z* 45 [C_2_H_7_N] when the isoquinoline ring was opened and the amino group along with two methyl groups were removed, this being an essential characteristic of aporphine alkaloids [14,15]. We then obtained a product ion at *m*/*z* 265.0852 (C_17_H_13_O_3_^+^) by the precursor-ion neutral loss of CH_3_OH. The presence of a fragment ion at *m*/*z* 282.0877 (C_17_H_14_O_4_^+^) manifested the parallel loss of CH_3_. The base peak for the fragment ions was obtained at *m*/*z* 265.0852, and the loss of CO was obtained at *m*/*z* 237.0910 (C_16_H_13_O_2_^+^).

Figure 2B shows the proposed fragmentation pathway for lirinidine. The neutral loss of CH_3_NH_2_ and the production of the ion at *m*/*z* 251.1067 (C_17_H_14_O_2_^+^), in addition to the consequent neutral loss of CH_3_OH and CO, yielded fragment ions at *m*/*z* 219.0806 (C_16_H_10_O^+^) and 191.0856 (C_15_H_10_^+^).

Figure 2D shows the proposed fragmentation pathway for roemerine. This yielded a fragment ion at *m*/*z* 249.0912 (C_17_H_13_O_2_^+^) because of the characteristic elimination of CH_3_NH_2_, and also involved the expulsion of CH_3_O, which produced a fragment ion at *m*/*z* 219.0805 (C_16_H_10_O^+^). The consequent neutral loss of CO generated a fragment ion at *m*/*z* 191.0856 (C_15_H_10_^+^).

Figure 2C shows the proposed fragmentation pathway for *N*-nornuciferine. The fragment ion at *m*/*z* 265.1224 (C_18_H_17_O_2_^+^) resulted from the characteristic elimination of NH_3_. Subsequently, the product ion at *m*/*z* 234.1041 (C_17_H_14_O^+^) was produced by the precursor-ion loss of OCH_3_. The fragment ion at *m*/*z* 250.0990 (C_17_H_14_O_2_^+^) was observed because of the parallel loss of CH_3_.

At *m*/*z* 45 [C_2_H_7_N], *m*/*z* 31 [CH_3_NH_2_], and 17 [NH_3_], the types of nitrogen substituted in aporphine alkaloids could be distinguished, representing quaternary, tertiary and secondary aporphine alkaloids, respectively; *m*/*z* 32 [CH_3_OH], 31 [CH_3_O], 28 [CO], and 18 [H_2_O] Da were all neutral loss fragments of aporphine alkaloids. At *m*/*z* 58 [C_3_H_8_N], four standards exhibited this characteristic peak in diagnosis fragmentation ions. As a result, the diagnostic product ion and the neutral loss are important in determining the fracture process for each chemical.

#### 2.2.1. Fragmentation Pattern for Quaternary Aporphine Alkaloids

We accurately identified Compound **27** as magnoflorine by comparing the retention time and the MS and MS^2^ spectra with the reference-standard data. We also found that Compounds **19** and **42** were eluted at 6.70 and 8.97 min, respectively, and they possessed the same MS^1^ at 342.1670 [M]^+^. They also exhibited five distinct fragment ion peaks at *m*/*z* 58.0658, 237.0905, 265.0852, 282.0877, and 297.1123. We identified these as magnoflorine isomers.

Compound **1** had a molecular formula of C_20_H_24_NO_5_ and a retention time of 3.32 min. This compound produced the precursor ion at 358.1649 [M]^+^ and four fragment ion peaks at *m*/*z* 58.0660, 227.0703, 255.0667, and 287.0917 in the positive ion mode. Based on secondary fragmentation data, we identified Compound **1** as C_6a_-hydroxylation of magnoflorine [16].

For Compounds **3**, **11**, **21**, and **29**, we determined a molecular design based on the structure of magnoflorine with one methoxy group removed, giving a molecular formula of C_19_H_22_NO_3_. We obtained the precursor ion at *m*/*z* 312.1594 [M]^+^ and observed four characteristic fragment ion peaks at *m*/*z* 58.0660, 217.0650, 207.0808, and 267.1018 in the positive ion mode. We tentatively identified these four compounds as C_2_-O-demethylation of magnoflorine isomers [16].

The isomeric Compounds **4**, **8**, **15**, and **35** exhibited identical fragment ions and molecular ions. The precursor ion at *m*/*z* 358.1649 [M]^+^ was formed using the chemical formula C_20_H_24_NO_5_. We observed five characteristic fragment ion peaks at *m*/*z* 58.0659, 253.0863, 281.0809, 285.0740, and 313.1071 in the positive ion mode. We identified these compounds as isomers of trilobinine [17].

Compounds **17** and **40** had a molecular formula of C_20_H_26_NO_3_ and produced a precursor ion at *m*/*z* 328.1907 [M]^+^ in the positive ion mode. We observed four characteristic fragment ion peaks at *m*/*z* 58.0659, 251.1067, 253.1226, and 283.1328. On the basis of the above information, we tentatively identified these compounds as isomers of *N*-ring-opening C_1_-dehydroxylation of magnoflorine [16].

Compound **16** had a chemical formula of C_20_H_26_NO_4_, was eluted at 6.22 min, and produced a precursor ion at *m*/*z* 344.1856 [M]^+^ in the positive ion mode. We observed five characteristic fragment ion peaks at *m*/*z* 58.0659, 137.0598, 143.0493, 175.0754, and 299.1278. On the basis of the above MS and previous findings in the literature, we identified this compound as zizyphusine+ 2H [18].

For Compounds **20**, **34**, and **46**, we designed a molecular structure from dihydroxylation of magnoflorine, giving it the molecular formula of C_20_H_24_NO_6_, and produced a precursor ion at *m*/*z* 374.1599 [M]^+^ in the positive ion mode. We observed three characteristic fragment ion peaks at *m*/*z* 58.0659, 297.0758, and 329.1022. We identified these compounds as isomers of di-hydroxylation of magnoflorine [19].

Compounds **22** and **26** had the chemical formula of C_21_H_28_NO_4_ and generated a precursor ion at *m*/*z* 358.20128 [M]^+^ in the positive ion mode. We observed three characteristic fragment ion peaks at *m*/*z* 58.0660, 281.0813, and 313.1446. We compared these data with previous findings in the literature and identified these compounds as isomers of pareirarinea [20].

Compounds **7**, **14**, **24**, and **39** had the formula C_20_H_22_NO_4_, with the same quasi- molecular ions [M]^+^ at *m/*z 340.1543 in the positive ion mode. We observed five characteristic fragment ion peaks at *m*/*z* 189.0692, 217.0644, 235.0754, 263.0703, and 295.0966. Drawing on the findings from previous research, we identified these compounds as isomers of *N*-methylbulbocapnine [21].

Compound **33** had a molecular formula of C_19_H_22_NO_5_ and a retention time of 8.04 min; it produced the precursor ion at *m*/*z* 344.1492 [M]^+^ in the positive ion mode. We observed four characteristic fragment ions at *m*/*z* 58.0660, 237.0907, 265.0860, and 283.0926. We tentatively identified Compound **33** as *N*-CH_3_-hydroxylation and C_2_-O-demethylation of magnoflorine [16].

Compounds **36**, **43**, and **49** were obtained form a molecular design in which one of the hydroxyl groups that is in magnoflorine becomes methoxy. These compounds had a molecular formula of C_21_H_26_NO_4_ and produced a precursor ion at *m*/*z* 356.1856 [M]^+^ in the positive ion mode. We observed characteristic fragment ion peaks at *m*/*z* 58.0660, 236.0833, 264.0785, 251.1066, 279.1018, 280.1082, 296.1038, and 311.1280. On the basis of the above molecular design and fragmentation information, we identified Compounds **36**, **43**, and **49** as menisperine isomers [22].

Compound **41** had a molecular formula of C_22_H_26_NO_6_ and a retention time of 8.83 min; it produced a precursor ion at *m*/*z* 400.1755 [M]^+^ in the positive ion mode. We observed four characteristic fragment ion peaks at *m*/*z* 58.0660, 295.0961, 323.0918, and 355.1180. We identified this compound as C_10_-OCH_3_-hydroxylation and C_11_-O-acetylation of magnoflorine [16].

Compounds **44** and **61** had a molecular design based on the structure of magnoflorine with one hydroxyl group and one methoxy group removed, giving a molecular formula of C_19_H_22_NO_2_, and it produced a precursor ion at *m*/*z* 296.1645 [M]^+^. We observed six characteristic fragment ion peaks at *m*/*z* 58.0660, 219.0807, 220.0842, 221.0957, 236.0826, and 251.1068 in the positive ion mode. On the basis of the above information, we identified Compounds **44** and **61** as isomers of C_1_-demethoxy-C_2_-dehydrox of magnoflorine [16].

Compound **48** had a molecular formula of C_22_H_26_NO_5_ and produced the precursor ion at *m*/*z* 384.18054 [M]^+^ in the positive ion mode. In addition, we observed seven characteristic fragment ion peaks at *m*/*z* 58.0659, 251.1067, 279.1019, 292.0738, 307.0953, 325.1070, and 339.1230. We therefore identified Compound **48** as C_1_- O-acetylation of magnoflorine [16].

The molecular design of Compound **52** was based on the structure of magnoflorine, from which one adjacent hydroxyl group and one methoxy group were removed, and one adjacent hydroxyl group and one methoxy group were changed to dioxolane, giving a molecular formula of C_19_H_20_NO_2_. This compound had a retention time of 13.07 min and produced a precursor ion at *m*/*z* 294.1489 [M]^+^. We observed four characteristic fragment ion peaks at *m*/*z* 58.0659, 191.0862, 219.0805, and 249.0911. On the basis of the above molecular design and fragmentation information, we identified Compound **52** as roemrefidine [23].

Compound **56** had a molecular formula of C_20_H_22_NO_5_ and a retention time of 13.53 min, and it produced a precursor ion at *m*/*z* 356.1492 [M]^+^ in the positive ion mode. We observed four characteristic fragment ion peaks at *m*/*z* 58.0659, 251.0703, 279.1028, and 311.0918. On the basis of the above fragmentation patterns, we tentatively identified Compound **56** as C_5_-methylene to ketone of magnoflorine [16].

For Compounds **59** and **64**, we obtained a molecular design based on the structure of magnoflorine, with an adjacent hydroxyl group in which one methoxy group was changed to 1,3-dioxolane and from which one methoxy group was removed, giving a molecular formula of C_19_H_20_NO_3_. These compounds had retention times of 14.11 and 17.08 min, respectively, and produced a precursor ion at *m*/*z* 310.1438 [M]^+^. We observed five characteristic fragment ions at *m*/*z* 58.0659, 177.0555, 205.0648, 233.0598, and 265.0859. On the basis of the above information, we identified Compounds **59** and **64** as isomers of C_1_-demethoxy -C_2_-dehydrox- C_10_, C_11_- ethyl epoxide of magnoflorine [16].

#### 2.2.2. Fragmentation Pattern of Tertiary Aporphine Alkaloid

We definitively identified Compound **45** as lirinidine by comparing its retention time and MS and MS^2^ spectra with reference standard data. Furthermore, Compounds **23** and **31** were eluted at 7.38 and 7.72 min, respectively, and exhibited the same MS^1^ at *m*/*z* 282.1489 [M+H]^+^. We observed five distinct fragment ion peaks at *m*/*z* 58.0660(90), 191.0855(5), 219.0806(23), 237.0911(100), and 251.1063, as with lirinidine. We therefore identified Compounds **23** and **31** as lirinidine isomers.

We precisely identified Compound **53** as roemerine by comparing its retention time and its MS and MS^2^ spectra with those in the reference standard data. 

Compound **5** had a molecular formula of C_18_H_15_NO_2_, was eluted at 5.33 min, and produced a precursor ion at *m*/*z* 278.1175 [M+H]^+^ in the positive ion mode. We observed three characteristic fragment ion peaks at *m*/*z* 107.0497, 246.0928, and 262.0858. We therefore identified this compound as dehydroroemerine [24].

For Compounds **6**, **12**, **18**, and **25**, the molecular design was based on the structure of lirinidine with an additional set of adjacent hydroxyls and methoxy groups, giving a molecular formula of C_19_H_22_NO_4_. These were eluted at 5.40, 5.88, 6.40, and 7.53 min, respectively, and they produced a precursor ion at *m*/*z* 328.1543 [M+H]^+^ in the positive ion mode. In addition, we observed five characteristic fragment ion peaks at *m*/*z* 58.0660, 177.0551, 222.1118, 265.0862, and 283.0967. On the basis of the above fragment characteristics, we identified these compounds as isomers of bolidine [25].

Compound **10** had a molecular formula of C_18_H_19_NO_4_, was eluted at 5.81 min, and produced a precursor ion at *m*/*z* 298.1437 [M+H]^+^ in the positive ion mode. We observed four characteristic fragment ion peaks at *m*/*z* 58.0659, 192.1022, 254.0953, and 283.1197. We identified Compound **10** as apoglaziovine [26].

Compounds **28**, **47**, and **62** had a molecular formula of C_19_H_19_NO_2_ and produced a precursor ion at *m*/*z* 294.1488 [M+H]^+^ in the positive ion mode. We observed six characteristic fragment ion peaks at *m*/*z* 58.0658, 217.0650, 236.0831, 250.0946, 263.1286, and 279.1256. On the basis of the above fragmentation patterns, we tentatively identified these compounds as dehydronuciferine isomers [27].

We produced Compound **30** by adding two adjacent methoxy groups and *N*-methyl to the structure of roemerine, giving a molecular formula of C_21_H_24_NO_4_. This compound exhibited a retention time of 7.69 min and produced a precursor ion at *m*/*z* 354.1670 [M+H]^+^. We observed three fragment ion peaks at *m*/*z* 58.0660, 251.1074, and 309.1119 in the positive ion mode. On the basis of these results, and previously reported findings in the literature [28], we identified Compound **30** as *N*-methylnantenine. 

Compounds **32**, **37**, and **58** had a chemical formula of C_19_H_21_NO_3_ and produced a precursor ion at *m*/*z* 358.20128 [M+H]^+^ in the positive ion mode. We observed five fragment ion peaks at *m*/*z* 58.0659, 217.0650, 267.1016, 280.1064, and 294.1487. On the basis of this information, we identified these compounds as isomers of isothebaine [29].

Compound **38** had a molecular formula of C_20_H_21_NO_4_ and exhibited a retention time of 8.66 min. This compound produced a precursor ion at *m*/*z* 340.1543 [M+H]^+^ in the positive ion mode. We observed four characteristic fragment ion peaks at *m*/*z* 58.0660, 220.0526, 264.0755, and 309.1354. Hence, Compound **38** was tentatively identified as crebanine [26].

Compounds **55**, **65** and **67** had a molecular formula of C_19_H_17_NO_4_ and produced a precursor ion at *m*/*z* 324.1230 [M+H]^+^ in the positive ion mode. We observed four characteristic fragment ion peaks at *m*/*z* 58.0659, 177.0554, 263.0940, and 293.1054. We identified these compounds as isomers of neolitsine [14].

Compounds **54**, **68**, and **69** had a molecular formula of C_18_H_13_NO_3_ and produced a precursor ion at *m*/*z* 292.0968 [M+H]^+^ in the positive ion mode. We observed three characteristic fragment ion peaks at *m*/*z* 248.0712, 264.1024, and 277.1039. The product ion at *m*/*z* 277.1039 [M+H−NH]^+^ was obtained via the neutral loss of NH. We ascribed the loss of this fragment to NH serving as a different substituent for nitrogen. On the basis of the above fragmentation patterns, we tentatively identified these compounds as lysicamine isomers [30].

Compound **60** had a molecular formula of C_19_H_19_NO_3_, exhibited a retention time of 14.22 min, and produced a precursor ion at *m*/*z* 310.1437 [M+H]^+^ in the positive ion mode. We observed three characteristic fragment ion peaks at *m*/*z* 58.0660, 279.1008, and 264.0792. On the basis of the above fragmentation patterns, we tentatively identified Compound **60** as stephanine [31].

Compound **63** had a molecular formula of C_18_H_15_NO_3_, exhibited a retention time of 15.99 min, and produced a precursor ion at *m*/*z* 294.1124 [M+H]^+^ in the positive ion mode. We observed four characteristic fragment ion peaks at *m*/*z* 58.0658, 239.0951, 257.1901, and 262.0863. On the basis of the above fragmentation patterns, we tentatively identified Compound **63** as *N*-formyl-annonain [32].

Compound **66** had a molecular formula of C_19_H_17_NO_3_, exhibited a retention time of 18.00 min, and produced a precursor ion at *m*/*z* 308.1281 [M+H]^+^ in the positive ion mode. We observed three characteristic fragment ion peaks at *m*/*z* 191.0859, 219.0806, and 249.0914. The product ion at *m*/*z* 219.0806 [M+ H−C_2_H_5_NO]^+^ was obtained via the neutral loss of C_2_H_5_NO. We ascribed the loss of this fragment to NHCOCH_3_ serving as a different substituent for nitrogen. Based on the secondary fragmentation data and mass spectral fragmentation behavior, we identified Compound **66** as *N*-acetylanonaine [33].

#### 2.2.3. Fragmentation Pattern of Secondary Aporphine Substituted

We unambiguously identified Compound **50** as *N*-nornuciferine by comparing its retention time and its MS and MS^2^ spectra with the reference standard data.

Compounds **2** and **13** had a molecular formula of C_25_H_31_NO_9_, and they produced a precursor ion at *m*/*z* 490.2072 [M+H]^+^ in the positive ion mode. We observed five characteristic fragment ion peaks at *m*/*z* 192.1019, 237.0901, 265.0861, 297.1122, and 328.1544. On the basis of the above fragments, we identified Compounds **2** and **13** as isomers of 11-glc- norisocorydine [18].

Compound **9** had a molecular formula of C_18_H_19_NO_4,_ was eluted at 5.77 min, and produced a precursor ion at *m*/*z* 314.1386 [M+H]^+^ in the positive ion mode. We observed six characteristic fragment ions at *m*/*z* 58.0660, 165.0913, 205.0658, 237.0910, 265.0861, and 297.1124. We identified this compound as laurolitsine [34].

Compound **51** had a molecular formula of C_17_H_15_NO_2_, exhibited a retention time of 12.80 min, and produced a precursor ion at *m*/*z* 266.1176 [M+H]^+^ in the positive ion mode. We observed four characteristic fragment ion peaks at *m*/*z* 131.0494, 191.0855, 219.0804, and 249.0912 in the positive ion mode. On the basis ofn the above information, we identified Compound **51** as anonaine [33].

Compound **57** had a molecular formula of C_18_H_17_NO_4_, was eluted at 13.83 min, and produced a precursor ion at *m*/*z* 312.1230 [M+H]^+^ in the positive ion mode. We observed five characteristic fragment ion peaks at *m*/*z* 58.0659, 264.1164, 265.0865, 280.1095, and 295.1328. We identified Compound **57** as nandigerine [21].

Compound **70** had a molecular formula of C_18_H_17_NO_4_, was eluted at 19.08 min, and produced a precursor ion at *m*/*z* 338.13860 [M+H]^+^ in the positive ion mode. We observed four characteristic fragment ion peaks at *m*/*z* 279.1258, 307.1201 308.1265, and 323.1153. We identified Compound **70** as sinomendine [35].

### 2.3. Pharmacological Activity of Aporphine Alkaloids in SSD

Natural aporphine alkaloids exhibit a wide range of biological properties, including antioxidant, antiplatelet-aggregation, anticonvulsant, antispasmodic, anti-cancer, antimalarial, antiprotozoal, anti-poliovirus, anticytotoxicity, and anti-Parkinson effects. Natural products and their synthetic derivatives from the mainstay of research can be made into new medications for a wide range of disorders [36].

Aporphine alkaloids are widely distributed in various medicinal plants and are the active ingredients in many traditional Chinese medicines. Magnoflorine is one of the most important pharmacologically active compounds in the quaternary aporphine alkaloid, with reported anti-diabetic and anti-inflammatory effects [37]. Lirinidine is a tertiary aporphine alkaloid which greatly inhibits the production of collagen and arachidonic acid and reduces the aggregation of platelet-activating factor-induced platelets [38]. Among the secondary aporphine alkaloids, norisocorydine can help regulate transporters in the small intestine [39] and *N*-nornuciferine exhibits anti-inflammatory effects [40].

## 3. Material and Methods

### 3.1. Chemicals and Materials

We obtained acetonitrile and LC grade methanol from MACKIN Company. We acquired MS grade formic acid from Thermo Fisher Scientific Co., Ltd. (New Jersey, NJ, USA). We obtained purified water from Guangzhou Watsons Food & Beverage Co., Ltd. (China). We acquired SSD samples from Liuzhi Special Zone, Liupanshui City, Guizhou Province, with an altitude of 1367M and a latitude and longitude 105°28′ E, 26°13′ N by Kunming Plant Biotechnology Co., Ltd. We obtained roemerine (purity ≥98%) and lirinidine (purity ≥98%) from Wuhan ChemFaces Biochemical Co., Ltd (Wuhan, China). We acquired *N*-nornuciferine from Chengdu HerbSubstance Co.,Ltd. Magnoflorine (purity ≥98%) from Sichuan Weiqi Biotechnology Co., Ltd. (Sichuan, China).

### 3.2. Standard and Solution Preparation

We pulverized an SSD stem and accurately weighed 1 g of sample powder. We transferred this to a flask containing 10 mL of 70% aqueous methanol (*v*/*v*) and performed ultrasonic extraction for 60 min at room temperature. We obtained supernatant after filtrating (nylon needle filter, 0.45 μm) and centrifuging at 13,523 g for 20 min at 10 °C. 

We prepared reference-standard stock solutions of magnoflorine, lirinidine, *N*-nornuciferine, and roemerine at concentrations of 0.1 mg/mL with methanol. These were stored at 4 °C.

### 3.3. Instruments and UHPLC-MS Conditions

We achieved a full characterization of aporphine alkaloid in SSD using a Dionex Ultimate 3000 RS UHPLC equipped with a quaternary pump and LPG-3400SD vacuum degasser unit (Thermo Fisher Scientific, California, CA, USA). We also used a Q-Exactive Orbitrap MS mass spectrometer with an electrospray ionization (ESI) source. We separated all the analyzed samples using a Thermo Scientific Hypersil GOLD™ aQ C18 column (100 mm × 2.1 mm, 1.9 μm) at 40 °C with a flow rate of 0.3 mL/min. The mobile phases consisted of 0.1% formic acid aqueous solution (solvent A) and acetonitrile (solvent B) with a flow rate of 0.30 mL/min. The gradient program was as follows: 0–2 min, 95–90% A; 2–5 min, 90–85% A; 5–10 min, 85–80% A; 10–12 min, 80–65% A; 12–20 min, 65–30% A; 20–22 min, 30–5% A; 22–22.1 min, 5–95%A; 22.1–25 min, 95%A, The sample injection volume was 2 μL. 

All samples were examined in the positive mode using the following tune approach. We used full-scan mode to produce high-resolution mass spectra with a resolution of 70 000 and a mass range of *m*/*z* 120–1000. PRM parameters were set as follows: the resolution was 35,000; the isolation window was 3.0 *m*/*z*; and the NEC (normalized collision energy) was set to 35, with 5.0× e^4^ of automatic gain control (AGC) target. We processed data using Xcalibur™ version 4.1 (Thermo Fisher Scientific, California, CA, USA) and Compound Discovery version 3.0 (Thermo Fisher Scientific, California, CA, USA). ESI source parameters were set as follows: the spray voltage was 3.5 kV; flow rates of 30 and 10 (arbitrary units) were used for the sheath gas and auxiliary gas, respectively; nitrogen was ≥99.99%; capillary temperature and the heater temperature were set to 320 °C and 350 °C, respectively; the S-lens RF level was 50.

### 3.4. Data Processing

We used the Thermo Xcalibur software version 4.1 and Compound Discover software version 3.0 (Thermo Fisher Scientific, California, CA, USA) to process all the raw data, including full-scan MS and MS^2^ data. We set the minimum peak intensity to 10,000 and calculated detailed chemical formula parameters from accurate masses for all the parent and fragment ions of selected peaks using a formula predictor, as follows: the maximum element counts were C30, H60, O20, and N10; the MS and MS^2^ mass tolerances were set to 5 and 10 ppm, respectively.

## 4. Conclusions

Using UHPLC Q-Exactive MS, we established an effective method to fully identify the aporphine alkaloids in SSD. We identified a total of 70 aporphine alkaloid constituents in SSD based on their chromatographic retention, MS and MS^2^, and bibliographic data. Sixty-nine of these are here reported as constituents of SSD for the first time. Some of these compounds have previously been shown to exhibit good pharmacological properties, including anti-cancer and anti-diabetic effects. Our findings lay the groundwork for more in-depth investigations of the pharmacodynamic substance basis for SSD.

## Figures and Tables

**Figure 1 molecules-27-07643-f001:**
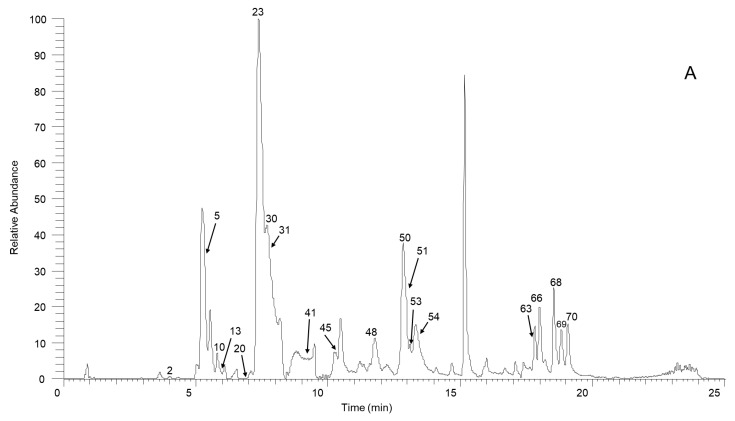
The high-resolution extracted ion chromatogram (HREIC) in 5 ppm for the multiple compounds in *Sabia schumanniana* Diels. (**A**) Peak 2, 13: *m*/*z* 490.2072; 5: *m*/*z* 278.1175; 10: *m*/*z* 298.1437; 20: *m*/*z* 374.1598; 23, 31, 45, 50: *m*/*z* 282.1489; 30: *m*/*z* 354.17; 41: *m*/*z* 400.1755; 48: *m*/*z* 384.1805; 51: *m*/*z* 266.1176; 53: *m*/*z* 280.1332; 54, 68, 69: *m*/*z* 292.0968; 63: *m*/*z* 294.1124; 66: *m*/*z* 308.1281; 70: *m*/*z* 338.1386; (**B**) Peak 3, 11, 21, 29, 32, 37, 58: *m*/*z* 312.1594; 6, 12, 18, 25: *m*/*z* 328.1543; 14: *m*/*z* 340.1543; 47, 62: *m*/*z* 294.1488; 57: *m*/*z* 312.1230; 59, 64: *m*/*z* 310.1438; 60: *m*/*z* 310.1437; (**C**) Peak 4: *m*/*z* 358.1649; 16: *m*/*z* 344.1856; 19, 27, 42: *m*/*z* 342.1700; 33: *m*/*z* 344.1492; 34: *m*/*z* 374.1598; 36, 43, 49: *m*/*z* 356.1856; 44, 61: *m*/*z* 296.1645; 55, 65, 67: *m*/*z* 324.1230; 56: *m*/*z* 356.1492; (**D**) Peak 7, 14, 24, 38, 39: *m*/*z* 340.1543; 9: *m*/*z* 314.1386; 52: *m*/*z* 294.1489; (**E**) Peak 1: *m*/*z* 358.1649; 8: *m*/*z* 358.1649; 15: *m*/*z* 358.1649; 17: *m*/*z* 328.1908; 22, 26: *m*/*z* 358.2013; 36: *m*/*z* 356.1856; 40: *m*/*z* 328.1907; 46: *m*/*z* 374.1598.

**Figure 2 molecules-27-07643-f002:**
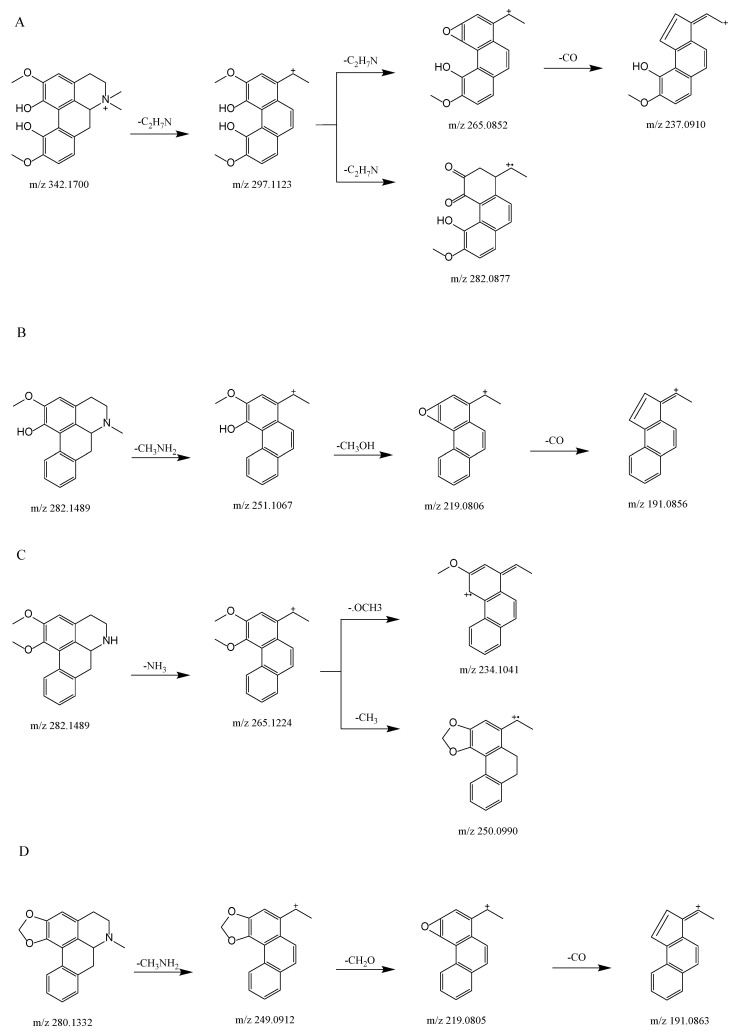
Proposed selected fragmentation pattern for components identified from SSD: magnoflorine (**A**); lirindine (**B**); *N*-nornuciferine (**C**); roemerine (**D**).

**Table 1 molecules-27-07643-t001:** The chromatographic and mass data for the components detected from *Sabia schumanniana* Diels though UHPLC-Q-Exactive Orbitrap MS.

Peak	t_R(min)_	Theoretical Mass *m*/*z*	Experimental Mass *m*/*z*	Error (ppm)	Formula (M+H)^+^ or (M)^+^	Identification
1	3.32	358.1649	358.1652	0.81	[C_20_H_24_NO_5_]^+^	C_6a_-hydroxylation of magnoflorine
2	4.02	490.2072	490.2079	1.43	[C_25_H_31_NO_9_+H]^+^	11-glc-norisocorydine isomer
3	4.05	312.1594	312.1600	1.86	[C_19_H_22_NO_3_]^+^	C_2_-O-demethylation of magnoflorine isomer
4	4.28	358.1649	358.1656	1.84	[C_20_H_24_NO_5_]^+^	trilobinine isomer
5	5.33	278.1175	278.1178	1.20	[C_18_H_15_NO_2_+H]^+^	dehydroroemerine
6	5.40	328.1543	328.1546	0.81	[C_19_H_22_NO_4_+H]^+^	bolidine isomer
7	5.71	340.1543	340.1552	2.66	[C_20_H_22_NO_4_]^+^	*N*-methylbulbocapnine isomer
8	5.72	358.1649	358.1651	0.56	[C_20_H_24_NO_5_]^+^	trilobinine isomer
9	5.77	314.1386	314.1389	0.75	[C_18_H_19_NO_4_+H]^+^	laurolitsine
10	5.81	298.1437	298.1440	0.70	[C_18_H_19_NO_3_+H]^+^	apoglaziovine
11	5.83	312.1594	312.1595	0.51	[C_19_H_22_NO_3_]^+^	C_2_-O-demethylation of magnoflorine isomer
12	5.88	328.1543	328.1545	0.35	[C_19_H_22_NO_4_+H]^+^	bolidine isomer
13	5.99	490.2072	490.2076	0.98	[C_25_H_31_NO_9_+H]^+^	11-glc-norisocorydine isomer
14	6.04	340.1543	340.1545	0.52	[C_20_H_22_NO_4_]^+^	*N*-methylbulbocapnine isomer
15	6.05	358.1649	358.1651	0.48	[C_20_H_24_NO_5_]^+^	trilobinine isomer
16	6.22	344.1856	344.1857	0.10	[C_20_H_26_NO_4_]^+^	zizyphusine+2H
17	6.30	328.1907	328.1908	0.15	[C_20_H_26_NO_3_]^+^	*N*-ring opening-C_1_-dehydroxylation of magnoflorine isomer
18	6.40	328.1543	328.1544	0.26	[C_19_H_22_NO_4_+H]^+^	bolidine isomer
19	6.70	342.1700	342.1702	0.54	[C_20_H_24_NO_4_]^+^	magnoflorine isomer
20	6.79	374.1598	374.1596	0.47	[C_20_H_24_NO_6_]^+^	di-hydroxylation of magnoflorine
21	7.04	312.1594	312.1597	1.09	[C_19_H_22_NO_3_]^+^	C_2_-O-demethylation of magnoflorine isomer
22	7.31	358.2013	358.2012	−0.32	[C_21_H_28_NO_4_]^+^	pareirarinea isomer
23	7.38	282.1489	282.1490	0.58	[C_18_H_19_NO_2_+H]^+^	lirinidine isomer
24	7.44	340.1543	340.1546	0.78	[C_20_H_22_NO_4_]^+^	*N*-methylbulbocapnine isomer
25	7.53	328.1543	328.1545	0.35	[C_19_H_22_NO_4_+H]^+^	bolidine isomer
26	7.58	358.2013	358.2008	1.41	[C_21_H_28_NO_4_]^+^	pareirarinea isomer
27 *	7.58	342.1700	342.1703	0.89	[C_20_H_24_NO_4_]^+^	magnoflorine
28	7.67	294.1488	294.1491	0.76	[C_19_H_19_NO_2_+H]^+^	dehydronuciferine isomer
29	7.68	312.1594	312.1597	0.90	[C_19_H_22_NO_3_]^+^	C_2_-O-demethylation of magnoflorine isomer
30	7.69	354.1700	354.1704	1.12	[C_21_H_24_NO_4_+H]^+^	*N*-methyl nantenine
31	7.72	282.1489	282.1491	0.80	[C_18_H_19_NO_2_+H]^+^	lirinidine isomer
32	7.87	312.1594	312.1597	0.90	[C_19_H_21_NO_3_+H]^+^	isothebaine isomer
33	8.04	344.1492	344.1495	0.85	[C_19_H_22_NO_5_]^+^	*N*-CH _3_ -hydroxylation of C_2_-O-demethylation of magnoflorine
34	8.04	374.1598	374.1603	1.49	[C_20_H_24_NO_6_]^+^	Di-hydroxylation of magnoflorine
35	8.10	358.1649	358.1653	1.23	[C_20_H_24_NO_5_]^+^	trilobinine isomer
36	8.35	356.1856	356.1858	0.52	[C_21_H_26_NO_4_]^+^	menisperine isomer
37	8.42	312.1594	312.1594	0.19	[C_19_H_21_NO_3_+H]^+^	isothebaine isomer
38	8.66	340.1543	340.1548	0.87	[C_20_H_21_NO_4_+H]^+^	crebanine
39	8.70	340.1543	340.1546	0.69	[C_20_H_22_NO_4_]^+^	*N*-methylbulbocapnine isomer
40	8.72	328.1907	328.1906	−0.24	[C_21_H_28_NO_4_]^+^	*N*-ring opening-C_1_-dehydroxylation of magnoflorine isomer
41	8.83	400.1755	400.1757	0.64	[C_22_H_26_NO_6_]^+^	C_10_-OCH_3_-hydroxylation and C_11_-O-acetylation of magnoflorine
42	8.97	342.1700	342.1702	0.63	[C_20_H_24_NO_4_]^+^	magnoflorine isomer
43	9.18	356.1856	356.1857	0.27	[C_21_H_26_NO_4_]^+^	menisperine isomer
44	9.71	296.1645	296.1646	0.45	[C_19_H_22_NO_2_]^+^	C_1_-demethoxy -C_2_-dehydrox of magnoflorine isomer
45 *	10.29	282.1489	282.1495	0.43	[C_18_H_19_NO_2_+H]^+^	lirinidine
46	10.32	374.1598	374.1599	0.34	[C_20_H_24_NO_6_]^+^	di-hydroxylation of magnoflorine
47	11.27	294.1488	294.1491	1.07	[C_19_H_19_NO_2_+H]^+^	dehydronuciferine isomer
48	11.76	384.1805	384.1812	1.64	[C_22_H_26_NO_5_]^+^	C_1_-O-acetylation of magnoflorine
49	12.82	356.1856	356.1861	1.39	[C_21_H_26_NO_4_]^+^	menisperine isomer
50 *	12.86	282.1489	282.1493	1.54	[C_18_H_19_NO_2_+H]^+^	*N*-nornuciferine
51	12.94	266.1176	266.1178	1.15	[C_17_H_15_NO_2_+H]^+^	anonaine
52	13.07	294.1489	294.1491	0.23	[C_19_H_20_NO_2_]^+^	roemrefidine
53 *	13.10	280.1332	280.1336	1.45	[C_18_H_18_NO_2_+H]^+^	roemerine
54	13.34	292.0968	292.0972	1.40	[C_18_H_13_NO_3_+H]^+^	lysicamine isomers
55	13.50	324.1230	324.1235	1.56	[C_19_H_17_NO_4_+H]^+^	neolitsine isomer
56	13.53	356.1492	356.1496	0.99	[C_20_H_22_NO_5_]^+^	C_5_-methylene to ketone of magnoflorine
57	13.83	312.1230	312.1231	0.33	[C_18_H_17_NO_4_+H]^+^	nandigerine
58	13.94	312.1594	312.1598	1.28	[C_19_H_21_NO_3_+H]^+^	isothebaine isomer
59	14.11	310.1438	310.1441	0.97	[C_19_H_20_NO_3_]^+^	C_1_-demethoxy -C_2_-dehydrox-C_10_,C_11_-Ethyl epoxide of magnoflorine isomer
60	14.22	310.1437	310.1440	0.87	[C_19_H_19_NO_3_+H]^+^	stephanine
61	14.26	296.1645	296.1649	1.20	[C_19_H_22_NO_2_]^+^	C_1_-demethoxy -C_2_-dehydrox of magnoflorine isomer
62	14.61	294.1488	294.1493	1.38	[C_19_H_19_NO_2_+H]^+^	dehydronuciferine isomer
63	15.99	294.1124	294.1125	0.07	[C_18_H_15_NO_3_+H]^+^	*N*-formyl-annonain
64	17.08	310.1438	310.1441	0.97	[C_19_H_20_NO_3_]^+^	C_1_-demethoxy -C_2_-dehydrox-C_10_,C_11_-Ethyl epoxide of magnoflorine isomer
65	17.39	324.1230	324.1231	0.33	[C_19_H_17_NO_4_+H]^+^	neolitsine isomer
67	18.18	324.1230	324.1234	1.16	[C_19_H_17_NO_4_+H]^+^	neolitsine isomer
68	18.53	292.0968	292.0972	1.12	[C_18_H_13_NO_3_ +H]^+^	lysicamine isomer
69	18.81	292.0968	292.0971	0.89	[C_18_H_13_NO_3_+H]^+^	lysicamine isomer
70	19.08	338.1386	338.1387	0.13	[C_20_H_19_NO_4_+H]^+^	sinomendine

* identified by comparison with standards.

## Data Availability

Data will be provided upon request.

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
