# Peer review of "A Systematic Method for the Identification of Aporphine Alkaloid Constituents in Sabia schumanniana Diels Using UHPLC-Q-Exactive Orbitrap/Mass Spectrometry"

_molecules, 2022, doi:10.3390/molecules27217643_

Round 1

Reviewer 1 Report

The authors identified numerous novel alkaloids using LC-MS based approach. In general, the work is interesting. The proposed fragmentation in figure 2 is commendable. Explaining fragmentation for the experimental compounds in sections 2.2.1 – 2.2.3 is nice, however, has a sense of monotony for the reader – it is very lengthy as well. Using diagnostic or qualifier fragment ions to explain the structure of related molecules is a great attempt however support of NMR data for some if not for all would be desirable, except when retention time is a match or when compared with respect to an authentic standard. Please also use the same font for the text and bibliography.

1.     Instead of listing fragment ions in table 1, I would suggest excluding that column from the table and out all the fragmentation spectra on a supplementary file.

2.     I don’t see the point of section 2.3. It’s from literature and just referencing this as part of the introduction would be better.

Author Response

please see to the attachment.

Reviewer 2 Report

I cannot recommend acceptance of the paper in the present state.

In general there is a lack of discussion of the results. The authors have limited themselves to presenting the results.

English language requires editing throughout manuscript.

Specific comments and suggestions are given in the manuscript (pdf) since there are no Line Numbers.

Reviewer 3 Report

English in the manuscript is quite bad (to the point of misunderstanding certain sentences) and definitely needs improvement (e.g. there is a typo in the title of the manuscript).

Names of compounds shouldn't be written with a capital first letter, unless they are commerical products (which they are not).

Abbreviations are used in the abstract without explanation - they are given repeatedly in the text, but the abstract is the first mention and one explanation in the first mention (i.e. in the abstract) should be enough.

PRM is a different term for MRM? Or is that a different method? It would be beneficial to the article to mentioned, whether tandem MS was used in the study.

The used HPLC mode should be mentioned in the abstract - based on the mobile phases described in 3.3 I assume reversed phases were used, but it would be useful to mention that earlier.

"The high resolution extracted ion mass spectrum of the SSD extract in the positive ion mode was illustrated in Figure 1" - the figure shows a chromatogram (ion current), not a spectrum!

Figure 1: It makes no sense to make the compounds with numbers from 1 to 67 and state the m/z values in the legend without assigning them to the numbers.

Figure 1: It is rather unusual to locate the compounds in different parts of a single unresolved peak (e.g. compounds 22, 23, 26 in Fig. 1a). In this case, I would consider different mode of presentation. Also I don't understand, on which criteria is based the division of chromatograms into A, B and C.

Table 1: The table should be adjusted so that the retention time values stay on one line. Current state is rather illegible.

Table 1: Names such as "C6a-Hydroxylation of magnoflorine" and "C2-O-Demethylation of magnoflorine" aren't names of compounds ("hydroxylation" and "demethylation" are processes, not substituents).

Table 1: Since the table is rather long, it would be beneficial to add the headline to each page (this might be handled better in the print version of the article, but should be of concern anyways).

Chapter 2.1: Selection of magnoflorine is not explained. I think it is the 70th - already known - compound found in SSD (apart from the 69 newly found), but this is just my deduction from the data and it is not stated explicitly soon enough (not until 2.2.1).

Chapter 2.1: I think I understand the logic behind the phrase "potential ions", but in my opinion, in English this means something else than what the authors aimed for and a different term should be used.

Figure 2: The text in the figure (e.g. atom names, m/z values etc.) should be significantly larger to enable comfortable reading. I will save any comments to the mechanisms for a revised version to be able to read them better.

Chapter 2.1: Thre is obviously a typo in stating that compound from Fig. 2C experiences a neutral loss of "CH3O".

The whole chapter 2.2 (with all of its subsections) should be shortened as it just basically repeats the data seen in Table 1. In my opinion, only interesting points regarding common losses and/or fragments should be mentioned (e.g. "loss of CH2NH2 group was observed in compounds 13-24 and 36-47" etc.).

Chapter 2.3: The second paragprah uses a different font. This will probably be handled in the print version, but I think it is still worth mentioning.

The citation format is inconsistent and also illegible.

Round 2

Reviewer 2 Report

The manuscript is improved.